# The Effects of Backfill Mining on Strata Movement Rule and Water Inrush: A Case Study

**Jian Hao [1,2], Yongkui Shi [1,\*], Jiahui Lin [1,\*], Xin Wang [1,\*] and Hongchun Xia [3]**

[1] State Key Laboratory of Mining Disaster Prevention and Control Co-founded by Shandong Province and the Ministry of Science and Technology, Shandong University of Science and Technology, Qingdao 266590, China; mkaqjs@163.com

[2] Shandong Tai'an Huaining Group, Taian 271400, China

[3] College of Civil and Architectural Engineering, Dalian University, Dalian 116622, China; xhch@dlu.edu.cn

[\*] Correspondence: shiyongkui@163.com (Y.S.); m17854258373_2@163.com (J.L.); woxingwangao@163.com (X.W.); Tel.: +86-178-542-58373 (J.L.)

**Abstract:** Backfill mining is widely used to control strata movement and improve the stress environment in China's coal mines. In the present study, the effects of backfill mining on strata movement and water inrush were studied based on a case study conducted in Caozhuang Coal Mine. The in-situ investigation measured abutment pressure distribution (APD), roof floor displacement (RFD), and vertical stress in the backfill area. Results are as follows: (i) The range and peak of APD, RFD, and vertical stress in the backfill area are smaller than in traditional longwall mining with the caving method. (ii) Backfill mining could change the movement form and amplitude of overburden and improve the ground pressure environment. (iii) Floor failure depth (FFD) is much smaller in backfill mining. Backfill mining can be an effective method for floor water inrush prevention.

**Keywords:** water inrush prevention; backfill mining; strata movement; ground pressure; floor failure depth

---

## 1. Introduction

In past decades, there has been a growing demand for coal resources in China. Hence, the depth at which coal deposits are extracted has increased considerably [1,2]. Deep coal mining has inherent risks associated with it due to in-situ stress and hydraulic damage conditions produced by overburden pressure, tectonic movements, and pressurized water in aquifers underlying the floor strata [3–5]. Historically, longwall mining with the caving method has been widely adopted in China. After removal of the mined panel, the overburden is divided into three zones: The caving zone, the fracture zone, and the slow subsidence zone [4,6]. These zones are demarcated according to differences in the magnitude of strata movement. Meanwhile, abutment pressure forms around the coal panel due to the movement of the overburden. Abutment pressure and strata movement may result in floor failure, which can induce water inrush.

Backfilling, an effective way to control strata movement induced by underground mining activities has been widely adopted in China [7–9]. In backfill mining, the goaf is filled with backfill materials, thus supporting part of the overburden stress and changing its movement rule and stress distribution. Therefore, there are great differences in strata movement and stress transfer between backfill mining and traditional longwall mining with the caving method. The problem of strata movement and stress transfer has been well studied in the past. Zhang et al. [10] described ground pressure behaviors in gauge backfill panels by theoretical analysis. Liu et al. [11] studied strata movement with a similar material experiment. Stresses in backfilled stopes were evaluated using numerical simulations [12,13].

However, only a few previous studies have investigated the deformation and stress response in the backfill area due to a lack of monitoring equipment.

To find out the strata movement rule and its influence on water inrush, in the present study, the surrounding rock's response to backfill mining was measured through an in-situ investigation conducted in Caozhuang Coal Mine. The backfill system, materials, and process were introduced. In field monitoring, multiple factors related with strata movement in backfill mining were considered, including abutment pressure distribution (APD), roof-floor displacement (RFD), vertical stress, and floor failure depth (FFD). Results indicate that backfill mining might improve the stress environment and decrease water inrush risks.

## 2. Study Area

The Feicheng mining area, located in Taian, Shandong Province, China, is famous for water inrush disasters and abundance of Ordovician limestone (Figures 1 and 2). Since the 1960s, 283 mine accidents have occurred in the Feicheng mining area. It is thought that 65% of recoverable reserves are lying above Ordovician water but are still seriously threatened by Ordovician water [14,15].

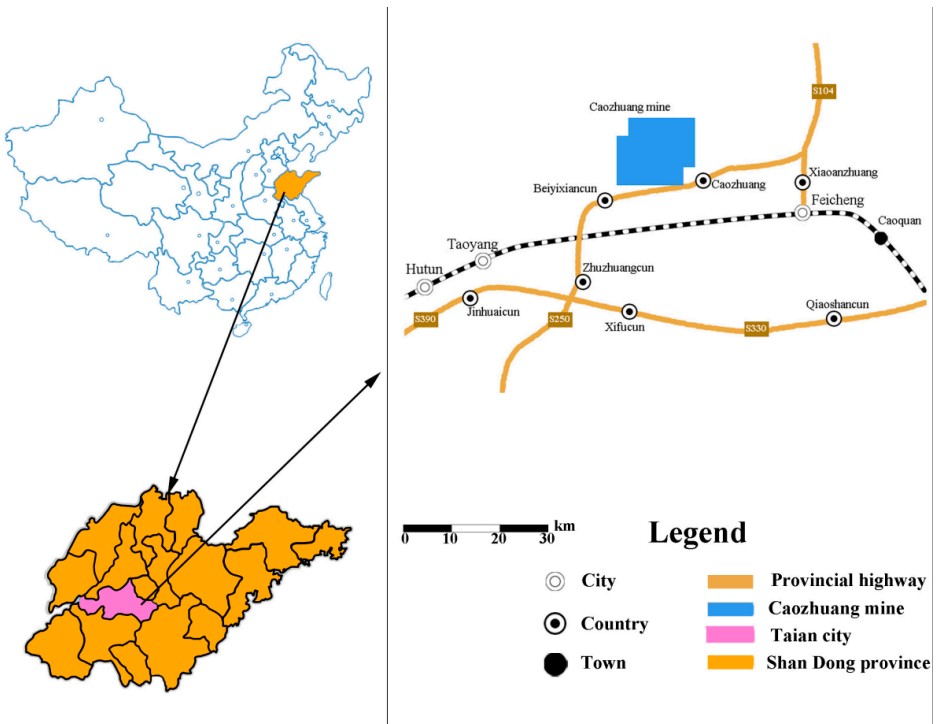

**Figure 1.** The location of Caozhuang Coal Mine.

The target coal seam is the #8 coal seam in Caozhuang Coal Mine CO., LTD of Feicheng Bureau of Shandong Energy Group CO., LTD. The #8 coal seam, excavated by coal panel 81006, has a depth of 550 m and belongs to the lower coal groups of the Northern China coal field. The coal seam has a mean thickness of 1.96 m, dips at an angle of 12–29°, and has a protodyakonov coefficient of 1.5. The immediate roof is the No. 4 limestone with a thickness of 5.3 m, and the floor is fine sandstone with a thickness of 4.78 m. The stratigraphic column of the mine field is shown in Figure 3. There are two water aquifers beneath the floor. The No. 5 limestone aquifer, with a bursting coefficient of 0.18 Mpa/m before mining, is 38 m away from the floor. The Ordovician limestone, with a bursting coefficient of 0.11 Mpa/m, is 61 m away from the floor. Additionally, faulted structures in the mine field are very complicated. The coal panel is severely threatened by the No. 5 and Ordovician limestone aquifers. Hence, the water-inrushing risks should not be overlooked.

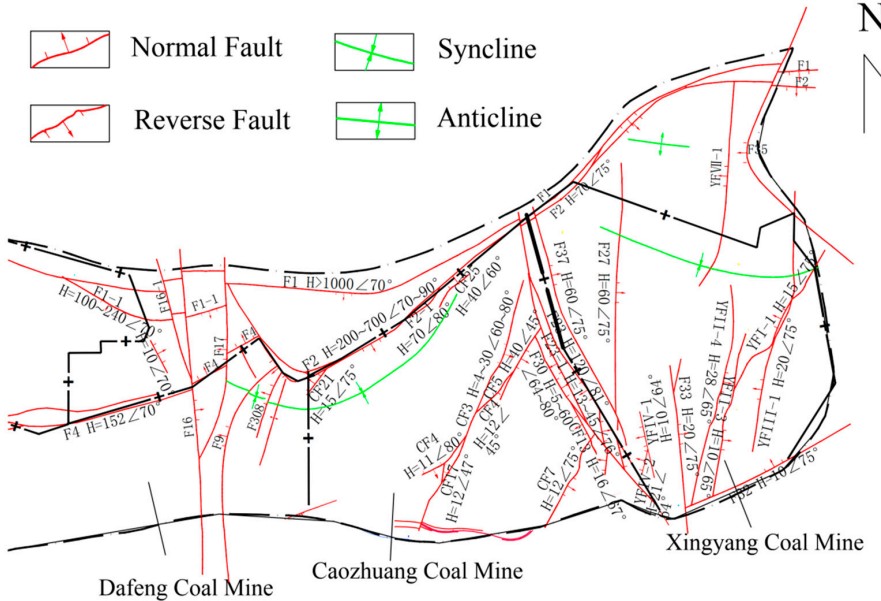

**Figure 2.** Geological map of Caozhuang Coal Mine.

| Lithology | Thickness(m) |
|---|---|
| No. 4 limestone | 5.3 |
| No. 8 coal | 1.4 |
| Fine sandstone | 4.68 |
| marlstone | 8.12 |
| No. 9 coal | 1.40 |
| Siltstone | 1.00 |
| No. 10-1 coal | 0.48 |
| Siltstone | 1.10 |
| No. 10-2 coal | 1.96 |
| Clay rock | 9.60 |
| Limestone | 1.50 |
| No. 11 coal | 0.46 |
| Fine sandstone | 7.60 |
| No. 5 limestone | 10.20 |
| Clay rock | 13.60 |
| Ordovician limestone | 800 |

**Figure 3.** Strata histogram of the mine field.

Water inrush disasters have had a great impact on coal extraction, and are not easily controlled by conventional water prevention and control methods. To prevent water inrushing disasters, non-pillar backfill mining is adopted in coal panel 81006 to pre-control the failure degree of the floor and prevent fracture due to water inrush. Longwall panels and backfill mining were designed respectively in coal panel 81006. For best performance of the grout-resistance wall and assuring successful backfill, the coal panel was divided into 4 sections from the inner to outer sections.

## 3. The Backfill System and Process

### 3.1. Materials and Methods

Paste-like backfill material is utilized in coal panel 81006 of Caozhuang Coal Mine. The material is comprised of cement, gangue powder, and coal ash, with a ratio of 1.8:6:6. The mass percentage of

the backfilling slurry is 60%, and the specific gravity is 1.56. The backfill uniaxial compressive strength with time is shown in Figure 4. The backfill material production process is shown in Figure 5.

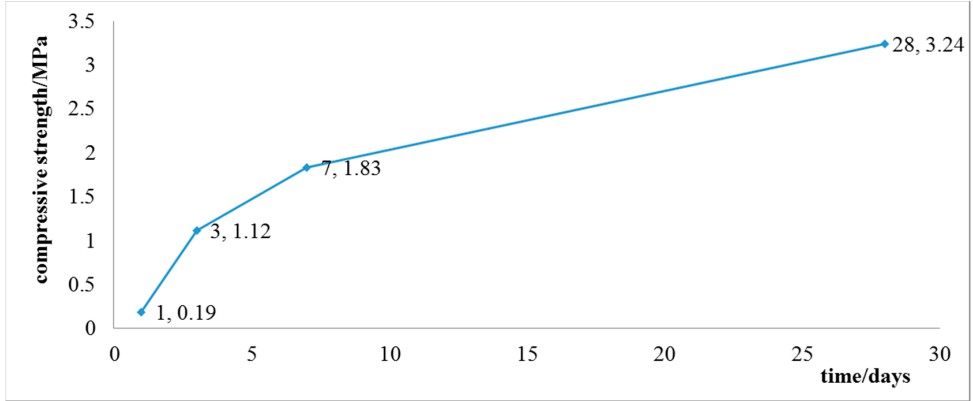

**Figure 4.** Backfill mass strength with time.

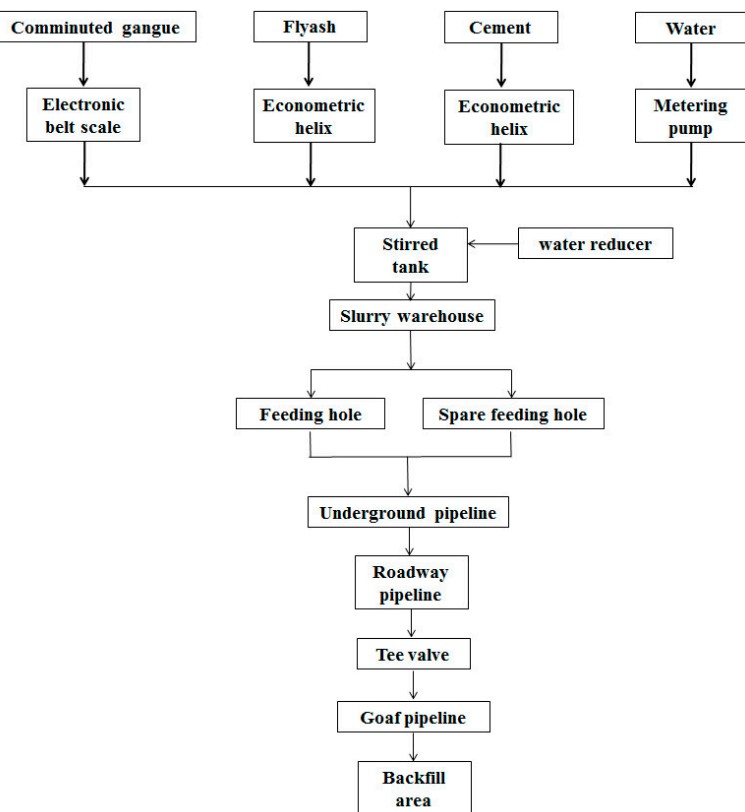

**Figure 5.** Diagram of the backfill material production process.

While the backfill system runs, the backfill slurry flows into the goaf behind the coal panels through feeding holes and a filling pipeline. The elevations of the feeding hole openings on the ground surface and the tail underground are +114.4 m and −240 m, respectively. The hole depth is 354.4 m. The elevation of the coal panel is −397.7 m to about −360 m, and the length of the backfill pipeline is 1850 m. The altitude difference from the feeding hole head to coal panels is 512 m, and the ratio of the pipeline length to altitude difference is 3.6. The diagram of the backfilling pipelines is shown in Figure 6.

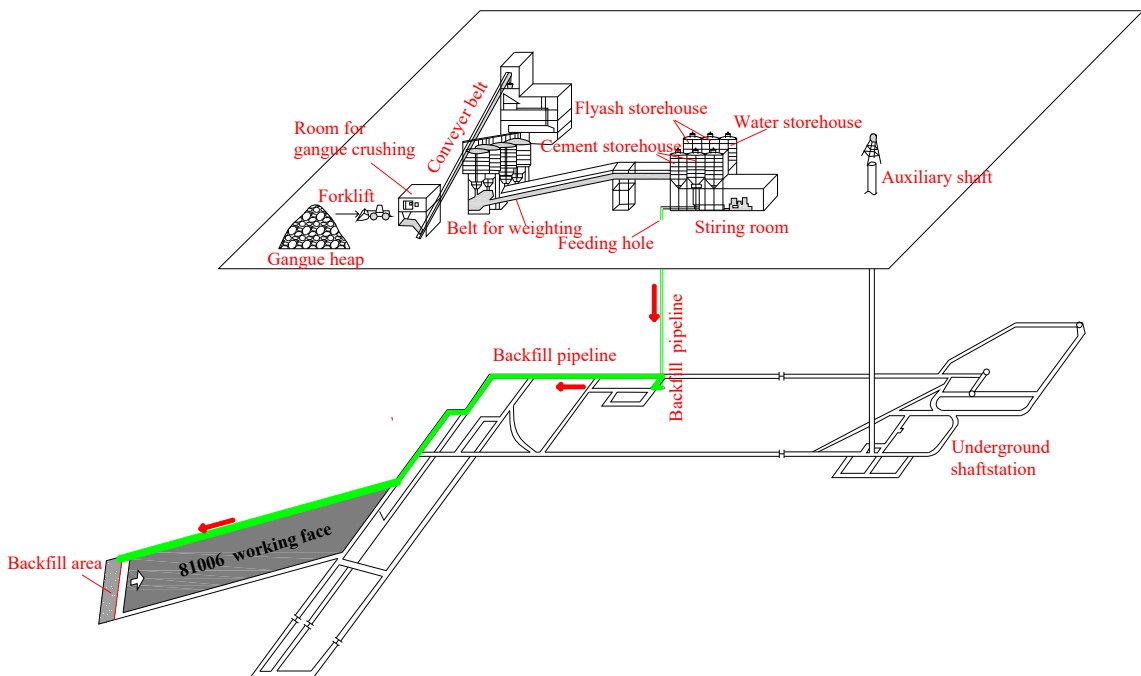

**Figure 6.** Diagram of backfilling pipelines.

## 3.2. Backfill System

The system consists of surface parts and underground parts. The backfill system has the characteristics of simple operation, real-time monitoring, tube-plugging reduction, and suitable transportation intensity. Material feeding, stirring, and transferring are controlled automatically in the backfill system. The surface system has 6 parts: The material production system, the material storage system, the water, and power supply system, the automatic control and metering system, the monitoring and communicating system, and the emergency system. The underground system has 2 parts, including the pipeline delivery and slurry-resistance systems in the coal panel. Upward mining was adopted in the longwall coal panel. Therefore, while the backfill system ran, the backfill material could flow down to the goaf. The coal panel layout is shown in Figure 7.

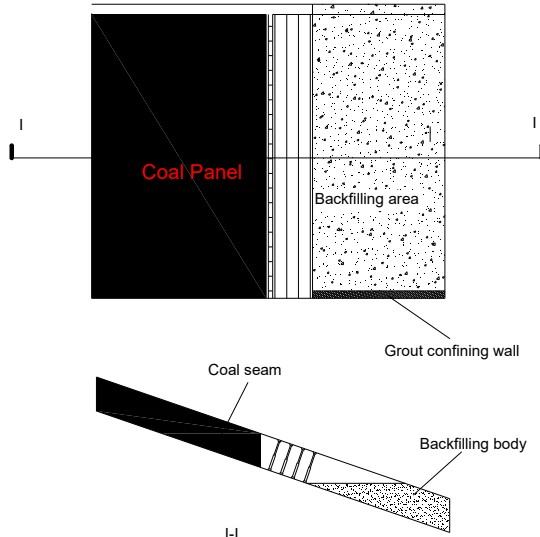

**Figure 7.** Overview of mining and filling in the coal panel.

## 4. Field Monitoring Equipment and Layout

### 4.1. Surrounding Rock Responding Monitoring Equipment in Backfill Coal Panel

#### 4.1.1. Introduction

To evaluate the effect of backfill mining, a dynamic monitoring system consisting of several meters was installed in the coal and backfill area. These meters were the MA15Z coal mass stress meter, MA650 roof displacement meter, and MA60 backfill stress meter (all shown in Figure 8), manufactured by Xian Xinyuan Company in China. Monitoring data consisted of APD beyond coal panels, the RFD inside backfill areas, and the vertical stress inside backfill areas. The MA15Z mine-specialized coal mass stress meters have a wide measuring range of 10 MPa and high accuracy of 0.01 MPa. They were able to monitor APD in front of the coal wall. For best performance, the MA15Z should be installed in a borehole ahead of the working face prior to mining.

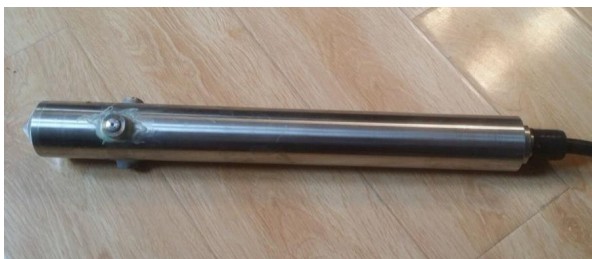

**(a)** The MA15Z mine-specialized coal stress meter.

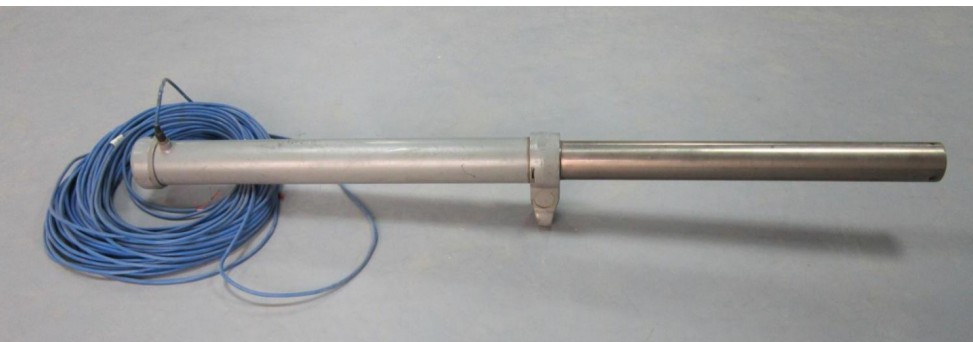

**(b**) The MA650 mine-specialized roof displacement meter.

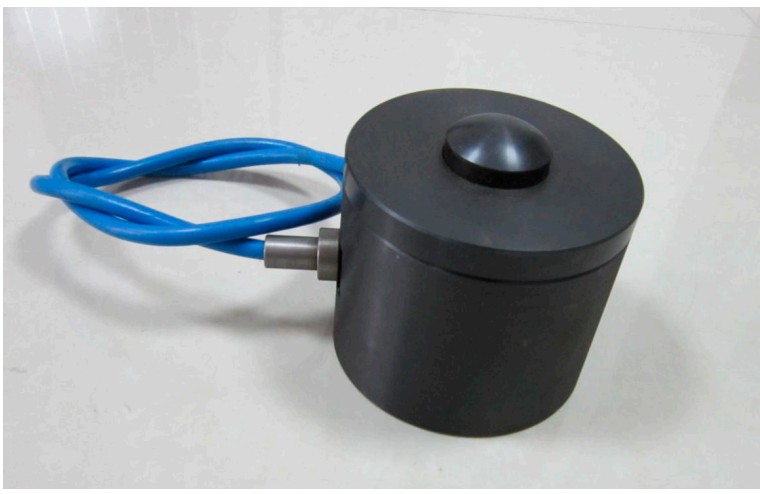

**(c**) The MA60 backfill stress meter.

**Figure 8.** The ground pressure monitoring sensor in backfill mining.

The MA650 roof displacement meter was installed in the backfill area and measured RFD there. As the roof compressed, the MA650's internal resistance bridge circuit calculated the compression and sent the data to a collector in real time. The MA60 backfill stress meter measured the vertical stress inside the backfill area and then sent data to a collector through communication lines.

### 4.1.2. Monitoring Equipment Layout

As the working face advanced, the monitoring system received data during the whole process. The MA15Z mine-specialized coal stress meters were installed into the coal mass before mining; the others were installed in the backfill area. The meters' layout is shown in Figure 9.

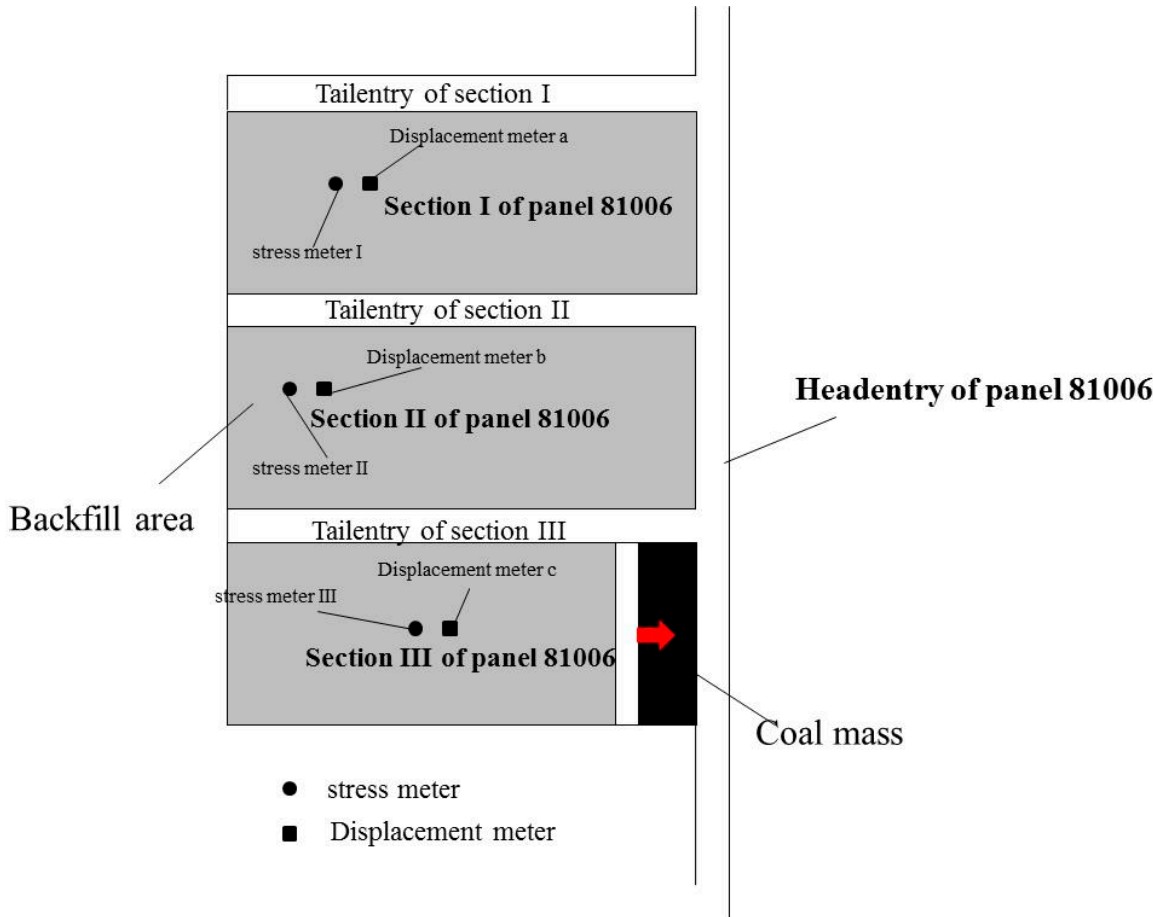

**Figure 9.** Meters' layout.

### 4.2. Floor failure Depth (FFD) Detecting Equipment and Detecting Method

### 4.2.1. Introduction

Isolated borehole section flow testing is used in many coal mines due to its simple construction/ installation and data analysis as well as low cost. The detection system consists of a testing device, a plugging control device, a water injection control device, and connector pipes as shown in Figure 10. It has two separate loops, including a plugging loop and a water injection loop. The packers at the ends of the testing device are used to isolate sections of the boreholes under the control of the plugging control device, and the leakage rate in the isolated section can be used to learn the failure condition of the strata.

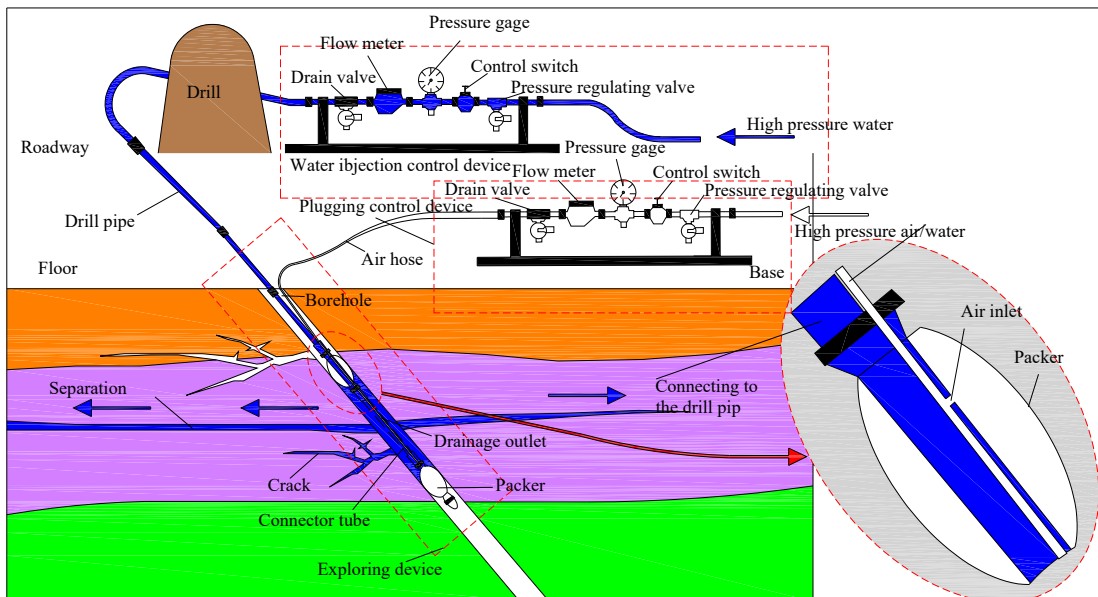

**Figure 10.** Diagram of an isolated borehole section flow testing system.

The exploring tube has two connected packers which stayed in a contracted state when not working. It can be pushed to any depth in a drill hole by the drilling tubes. By injecting air into the packers through a pressure regulating valve, the packers expanded into a spherical shape and formed an embolism at both ends of the exploring tube. A borehole blockage about 1m long then formed. Water was injected at a constant pressure through the drilling tubes, the regulating valve, and the pressure gauge. Then, the volume of water leakage per unit time through fractures in the hole wall was measured.

Theoretically, it can be proven that under a certain water injection pressure, the water injection flow is determined by the permeability of the rocks and the size of the fractures. Thus, the water-injection flow should increase with the increasing permeability coefficient and growth of the fractures. Results from the actual measurement demonstrated that the water-injection flow value is less than 1 L/min and even approaches 0 in unbroken rocks with a water-injection pressure of 0.1 MPa/min in every 1 m hole segment. In highly fractured rock, the flow value can be up to 30 L/min.

### 4.2.2. Detecting Method

In order to probe floor failure depth (FFD) of coal panel 81006, two drilling areas (A and B) were designed in the tail entry of section II. Drilling area A contains probing holes $D_1$ and $D_2$. Hole $D_1$ had an azimuth angle of 321°, a dip angle of −1°, and a depth of 91.8 m. Hole $D_2$ had an azimuth angle of 321°, dip angle of −8°, and depth of 77 m. Drilling area B contains probing holes $D_3$ and $D_4$. Hole $D_3$ had an azimuth angle of d 240°, a dip angle of 3°, and depth of 91.8 m. Hole $D_4$ has an azimuth angle of 240°, a dip angle of −3°, and depth of 87 m.

## 5. Results and Discussion

### 5.1. Abutment Pressure Distribution Characteristics

A coal stress meter was installed in the tail entry of section 1 of panel 81006 in front of the working face. A hole was drilled in the coal face and the meter was injected into it prior to mining. These meters monitored APD as the working face advanced. The coal stress meters monitored the variation of relative stress inside the coal mass which reflected the APD as the working face advanced. The monitoring results are shown in Figure 11.

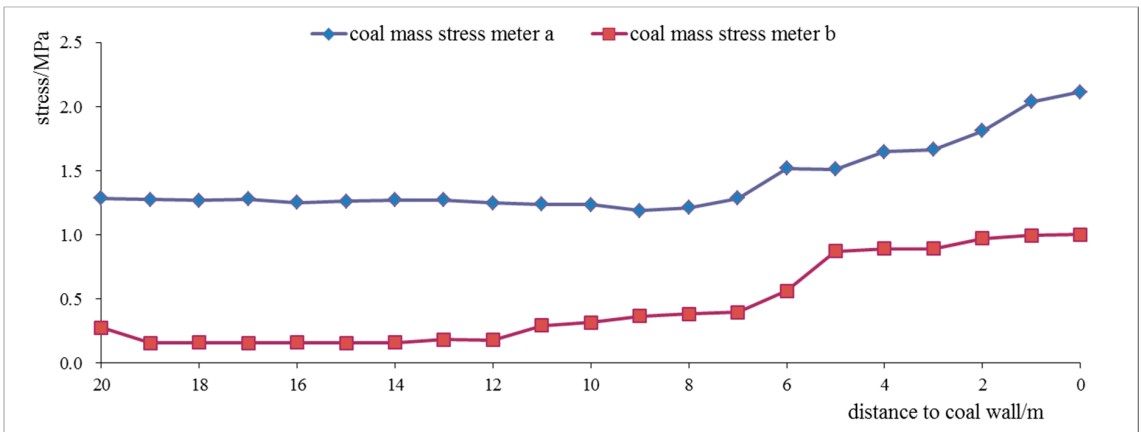

**Figure 11.** Variations of coal stress in the meters.

Figure 11 shows the relative stress in the coal mass as the working face advanced. The horizontal axis shows the distance from the installation location of the stress meter to the coal wall. As the working face advances, the relative stress in the coal mass increased gradually until reaching a maximum when the working face arrives at the installation location. It can be inferred that APD range, with no plastic zone, is approximately 12 m. Abutment pressure peaks at the wall edge and declines from there, indicating that abutment pressure concentration factor was small and the coal mass beyond the working face is in an elastic state.

The buried depth of the 81006 coal panel is about 550 m, and the uniaxial compressive strength of the coal is 15 MPa as provided by the mine. The abutment pressure concentration factor would be 2–3 normally [16–18]. We selected 2. If traditional longwall mining technology were adopted, the maximum supporting pressure would be as follows:

$$2\gamma H = 2.0 \times 2.5 \text{ t/m}^3 \times 550 \text{ m} = 27.5 \text{ MPa} \tag{1}$$

where $\gamma$ is the strata's average density, 2.5 t/m$^3$ and $H$ is the coal body's buried depth, 550 m.

Through theoretical analysis, the abutment pressure's peak in a traditional working face is higher than the uniaxial compressive strength of the coal body, creating a plastic zone at the coal wall edge. Monitoring data in other coal panels of this coal mine could prove this point. In backfill mining, according to the monitoring data, there is no plastic zone in front of the working face. This is because the backfill area supports the overburden, so less ground pressure is transferred to the coal mass in front of the working face. The ground pressure near the working face is therefore smaller.

*5.2. Roof-Floor Displacement (RFD) Monitoring in the Backfill Area*

The displacement meter monitored the amount of RFD in the backfill area, which reflected the deformation rule and the backfill compression ratio underground pressure. These two parameters are important to evaluate the backfill effect. The graph in Figure 12 was drawn according to the data gathered from the backfill displacement meter. From this graph, the greatest RFD of displacement meter was 104 mm. The curve's slope is relatively high at first, which indicates that the displacement between the roof and the floor happens quickly; then the slope decreases gradually, and the roof-floor displacement also tends to be stable and slow as the roof and floor reaches stable conditions.

It took about 70 days until the roof and the floor were stable, and the greatest RFD was 104 mm. The mining height is calculated as 2 m and the compression ratio of the backfill is 5.2%. The greatest RFDs at b and c are 93 mm and 129 mm, and the compression ratios are 4.74% and 6.45%, respectively. The curve variation rules of b and c are basically the same as that of displacement meter a.

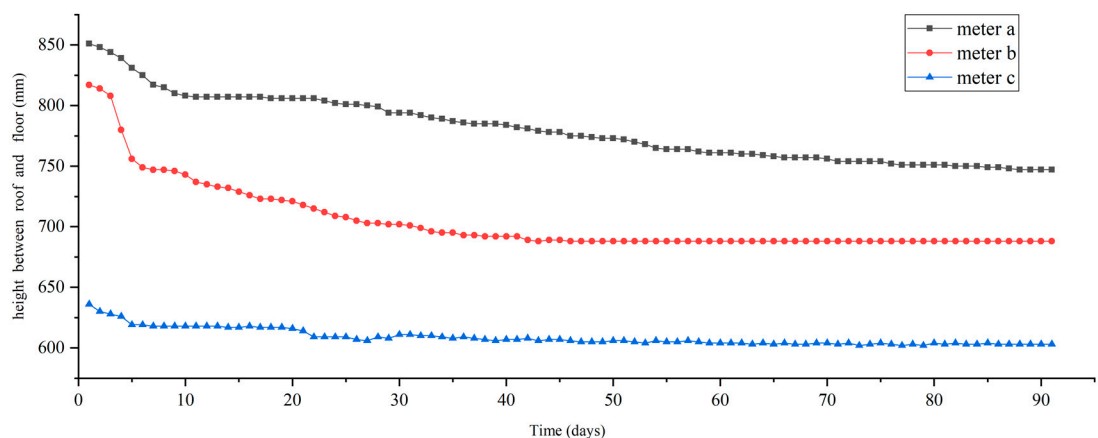

**Figure 12.** Variation of displacement in meter a.

### 5.3. Variations of Vertical Stress in the Backfill Area

Inside the backfill area, several backfill stress meters were installed to monitor the vertical stress variations with the advancement of the working face. As shown in Figure 13, the curve graph was drawn based on the data obtained by monitoring.

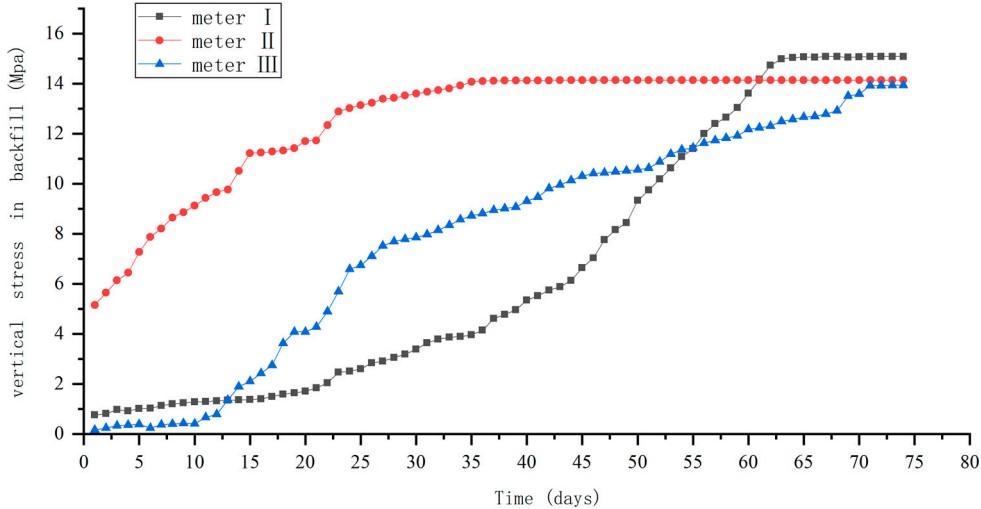

**Figure 13.** Variation of stress meter I.

From Figure 13, initially, ground pressure transferred to the backfill area is low and increases slowly. Then, vertical stress in the backfill area increases gradually until it reaches stability. The final stress value is 15.06 MPa, almost equal to the initial vertical stress ($\gamma H$ = 13.75 MPa), indicating that the backfill area has adequate capacity to support the overlying strata. This is also why the abutment pressure distribution's range is relatively narrow. It took around 70 days for vertical stress to stabilize, consistent with the displacement monitoring result.

Backfill stress monitored by meters II and III stabilized at 13.02 MPa (data before transmission line was destroyed) and 13.92 MPa, respectively. Their stress histories are shown in Figures 14 and 15, which had much of the same variation rule as stress meter I.

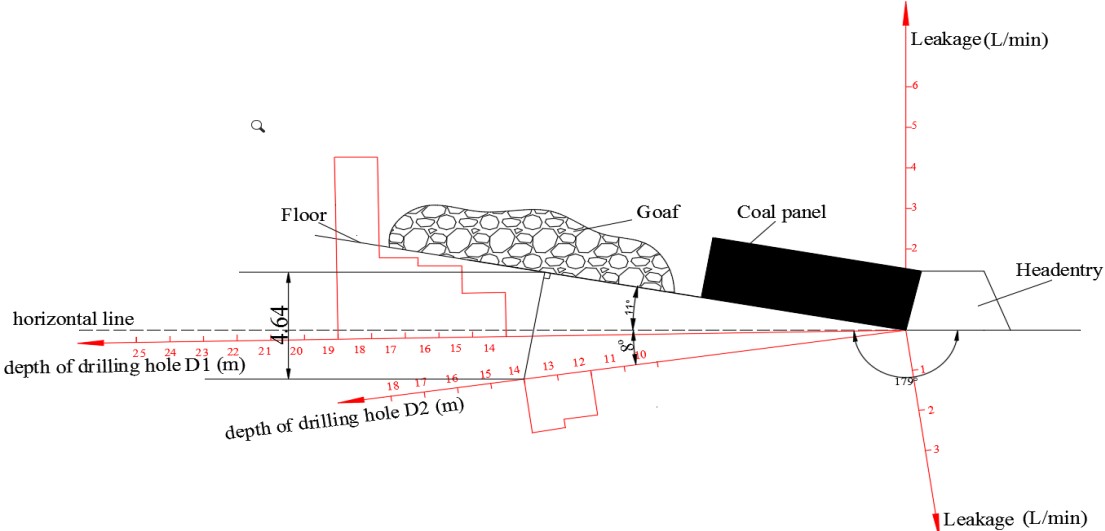

**Figure 14.** The leakage rates of drilling holes d1 and d2.

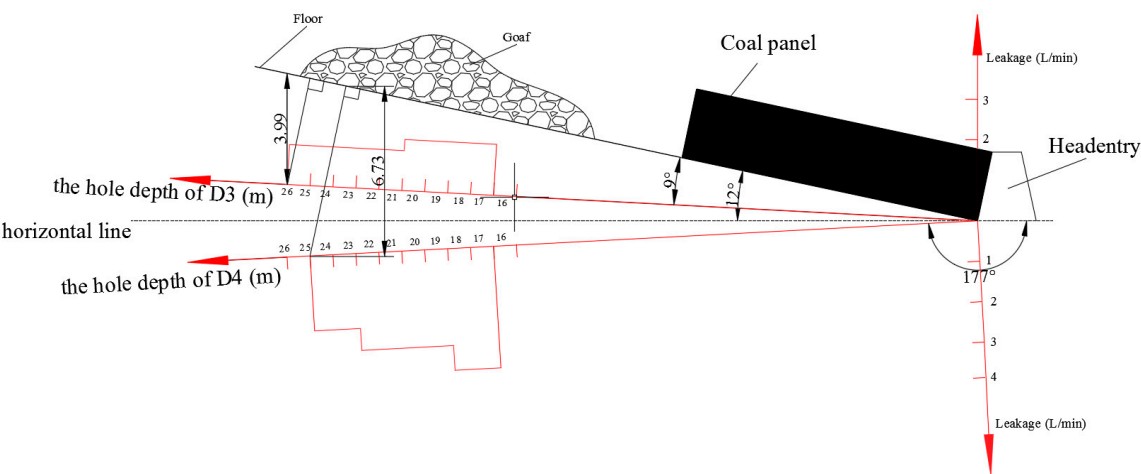

**Figure 15.** The leakage rates of drilling holes d3 and d4.

### 5.4. FFD in Backfill Mining

According to the probing data, the graph of the leakage rate of all drill holes was drawn as shown in Figure 14. From Figure 14, drill hole $D_1$ has a certain leakage amount from depths of 14–19 m. The leakage rate of other segments in $D_1$ is zero, indicating that rock fractures developed in this segment, which is the floor failure zone. Drill hole $D_2$ has a certain leakage amount from depths of 12–14 m, indicating that rock fractures developed in this segment, which is the floor failure zone. Analyzing the probing results of the two drill holes, the FFD is 4.64 m. From Figure 15, $D_1$ has a certain leakage amount above a depth of 25 m, while the leakage amounts of other segments are zero, indicating that rock fractures developed in this segment. $D_2$ has a certain amount of leakage above a depth of 27 m, indicating that rock fractures developed in this segment. Therefore, after analyzing the probing results of the two drill holes, the greatest FFD is 6.73 m.

### 5.5. Differences Between Strata Movement Rule of Backfill Mining and Traditional Longwall Mining and its Effect on Water Inrush

According to the above-mentioned field investigations, the abutment pressure did not surpass the strength limit of the coal, which remained in an elastic state. The range and peak of abutment pressures are relatively small compared to traditional longwall mining. The abutment pressure caused by strata movement was an important power resource to surrounding rock failure, and also to floor

failure depth. Obviously, backfill mining could change the abutment pressure range and intensity. Therefore, through backfilling mining, the stress environment can be improved, and surrounding rock failure could be reduced to some extent.

Vertical stress in the backfill area reached the original vertical stress computed by theoretical analysis, indicating that the weight of the overburden could be transferred to the backfill area after the working face moved forward a certain distance. The weight of the overburden will never disappear, but it will transfer to the surrounding rock. When vertical stress in the goaf is higher, stress in the surrounding rock (reflected by abutment pressure) will be lower. When the stress in the backfill approaches the original vertical stress, there will be little weight transfer to surrounding rock; therefore, the abutment pressure will be smaller.

RFD values in the backfill area of coal panel 81006 are 104 mm, 93 mm, and 129 mm, respectively, for sections I, II and III. The compression ratios are 5.2%, 4.74%, and 6.45%, respectively. We can infer that in backfill mining, RFD is much smaller than with the traditional caving method. The form of roof movement was changed in backfill mining. The roof did not cave, but rather it bent and subsided.

From the comprehensive analysis of the above characteristics, it is concluded that backfill mining with the paste-like material is quite different from traditional longwall mining. In traditional longwall mining, coal extraction results in the lower strata losing their original balanced state, bending, falling, and then creating rooms for the upper strata's motion. As the coal face advances, the overburden moves upward from the bottom. According to the differences in the magnitude of strata movement, the overburden is normally divided into three zones: The caved zone, fractured zone, and continuous bending zone (abbreviated as "three zones") [16,17,19], as shown in Figure 16. As the working face advances, the overburden's weight that is no longer supported by coal mass due to coal extraction moves to the surrounding rock in which the concentrated stress, called abutment pressure, forms. Generally, the concentrating coefficient K of the abutment pressure can reach 2–3, thus forming the plastic zone in front of the coal face.

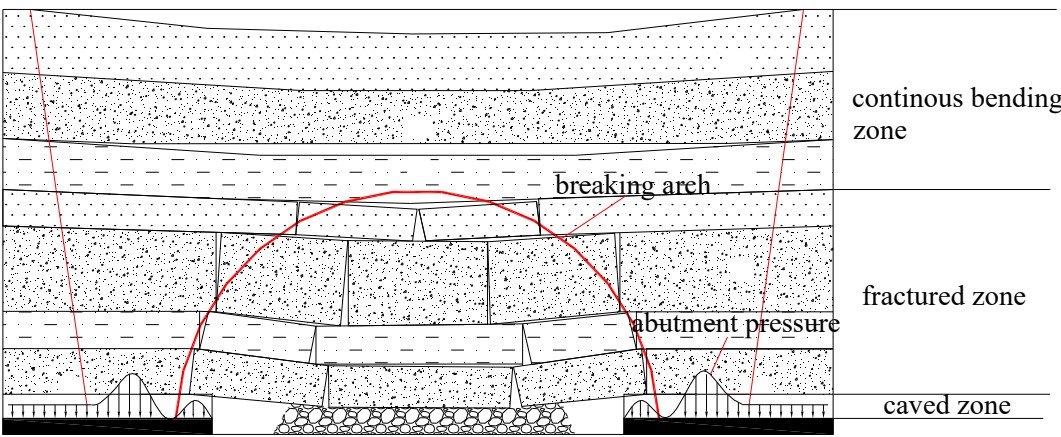

**Figure 16.** Strata movement characteristics in longwall mining.

In backfill mining, the goaf is backfilled before the immediate roof caves, leaving little room for the overburden's movement. The empty space that causes caving and subsidence no longer exists, which differs from longwall mining [8–10]. Due to the supporting effect of the backfill area, the lower strata obtain a new balanced state after the coal is replaced. The immediate roof does not cave or break. Furthermore, most of the overburden's weight is transferred to the backfill area. High-stress concentrations do not form in the coal. Commonly, no plastic zone appears in front of the coal wall. The abutment pressure peaks at the coal wall's edge. The concentration coefficient *K* of the abutment pressure in backfill mining is lower than that in traditional longwall mining, as shown in Figure 17.

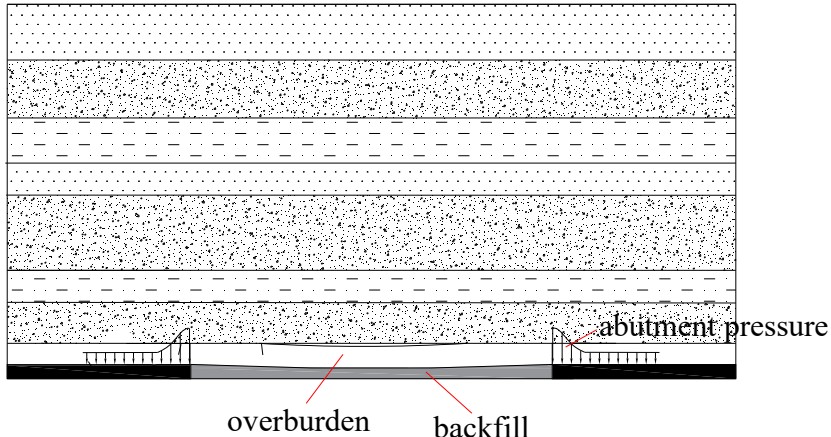

**Figure 17.** A structural model of strata movement in backfill mining.

In summary, the major differences between traditional longwall mining and backfill mining are given in Table 1.

**Table 1.** Strata movement characteristics of longwall caving mining and backfill mining.

| Serial Number | Index | Traditional Longwall Mining | Filling Mining |
|---|---|---|---|
| 1 | Overburdens movement scope | "Three zones" exist | "Three zones" do not exist |
| 2 | Overburdens movement form | Immediate roof caves | Immediate roof does not cave |
| 3 | Roof subsidence | Great | Small |
| 4 | Abutment pressure | High | Low |
| 5 | Plastic zone | Wide plastic zone | Narrow or no plastic zone |

Cracks in a floor disturbed by mining could provide a water inrush channel, and, therefore, FFD is an important index by which to evaluate coal panel safety. The greater is the FFD, the more dangerous the threat of a water inrush disaster. The FFD has a great relationship with strata movement and ground pressure, which can be reflected by APD, RFD, and vertical stress in backfill. The monitoring mentioned above showed that APD, RFD, and vertical stress in backfill are much smaller than in traditional longwall mining, implying that the form and aptitude of strata movement and ground pressure were improved significantly. It was not difficult to deduce that FFD would be relatively small, which may have an influence on water inrush disaster to some extent.

Monitoring results showed that the greatest FFD in the 81006 coal panel was 6.7 m, which agrees with our deduction above. According to previous monitoring data in other coal panels of the same coal seam extracted by traditional longwall mining, the FFD is more than 20 m. Thus, FFD in backfill mining is much smaller than in traditional longwall mining. Backfill mining can prevent water inrush channels from forming and developing and may be useful for water inrush prevention especially in mines with high floor pressure water.

## 6. Conclusions

A case study on a trial test of water inrush prevention technology with backfill mining was carried out in Caozhuang Coal Mine. According to field observations and monitoring, the trial test showed good control of ground pressure and floor fracture development by employing backfill mining.

(1) In-situ measurements of abutment pressure show that its range is about 10 m, and it peaks at the edge of the coal wall. Compared to traditional longwall mining, the range and peak of the abutment pressure with backfill technology are much smaller. Abutment pressure caused by strata movement was an important power resource to surrounding rock failure and to floor failure depth. Obviously, backfill mining could change the abutment pressure range and intensity, and therefore,

the stress environment of surrounding rock was improved implying that surrounding rock failure could be decreased to some extent, which can decrease the danger of water inrush.

(2) RFDs monitored in the backfill area of coal panel 81006 are 104 mm, 93 mm, and 129 mm, and the compression ratios are 5.2%, 4.74%, and 6.45%. Roof displacement with backfill technology is much smaller than with the traditional caving method. Roof displacement could reflect strata movement intensity, and roof displacement is very small without caving in backfill mining, indicating that backfill mining can change the form of the strata movement.

(3) Vertical stresses monitored in backfill areas I, II, and III were respectively 15.06 MPa, 13.02 Mpa, and 13.92 Mpa. These values were roughly equal to the original vertical stress (13.75 MPa), indicating that the weight of the overburden has been transferred to the backfill area. The weight of the overburden will never disappear, but it will transfer to the surrounding rock. When vertical stress in the goaf is higher, stress in surrounding rock (reflected by abutment pressure) will be lower. When the stress in backfill approaches the original vertical stress, there will be little weight transfer to surrounding rock, and therefore, the abutment pressure will be smaller. The results of vertical stress and abutment pressure indicated mutual verification.

(4) The case study indicates that backfill mining could change the form and aptitude of strata movement. Meanwhile, backfill mining could improve the stress environment, which could decrease FFD. The "three zones" do not appear in backfill mining. Rather, the ground pressure is much smaller with backfill technology, and FFD is much smaller in backfill mining, implying that backfill technology can be helpful for preventing floor water inrush. This study demonstrated that through backfill mining, FFD can be significantly decreased, and water inrush disasters may be eliminated. Backfilling mining may be an optional method for floor water inrush prevention and control.

**Author Contributions:** Conceptualization, Y.S.; Data curation, Y.S., X.W. and H.X.; Formal analysis, J.H.; Investigation, J.H. and J.L.; Methodology, H.X.; Resources, X.W.; Supervision, J.H.; Visualization, J.L.; Writing—review & editing, J.H.

**Funding:** This study was supported by the National Natural Science Foundation of China (Grant No.51804180, No.51574055), the key research plan of Shandong province (Grant No.2018GSF116007), the Natural Science Foundation of Shandong province (Grant No. ZR2017BEE033), Postdoctoral Innovation Foundation of Shandong province (201703076), "Top Disciplines" Special Project of Mining and Safety Engineering College(1SY04602).

**Conflicts of Interest:** The authors declare no conflict of interests.

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
