# Peer review of "The Effects of Backfill Mining on Strata Movement Rule and Water Inrush: A Case Study"

_processes, doi:10.3390/pr7020066_

Round 1

Reviewer 1 Report

Dear Authors and Editor,

Thanks for your kind invitation to review the manuscript entitled “The effects of backfill mining on strata movement rule and water inrush: a case study” by Jian Hao, Yongkui Shi, Jiahui Lin, Xin Wang and Hongchun Xia.

This document investigates the effects of backfill mining on the deformation of the rock massif and on the water influx through the analysis of pressures, tensions, displacements and deformation of the rock mass in the area of filling. It is concluded that the backfill mining is an effective procedure to improve the deformation of the rock and, in turn, the water inflow into the galleries filled with sterile.

In my opinion the article is well structured and can be published in this journal, however I think that some concerns should be taken into account through a Mayor Revision. Here are my suggestions:

1. To guide the reader, in the final part of the introduction should be explicitly stated and more clearly what are the objectives of the work. Next, a brief description of the content of the article, its parts or sections with a brief description should be made.

2. In the study area section the geological structure in the immediate vicinity of the mine is clearly described, but I believe that a more general vision should be provided, at a regional scale, including geological map with stratigraphic units or geological formations, tectonic structure with structural directions and dips, and lithological description. A cross section on a regional scale should also accompany this figure.

3. In general, the work is very focused as a case study. In my opinion, it should be given a broader meaning. For example, in Results and discussion section, apart to commenting on the specific quantitative data for the Caozhuang Coal Mine, more general, qualitative comments should be made and the repercussions that may be useful for places other than this mine should be commented.

4. With the conclusions the same thing happens, in my opinion they are too poor since they are limited to giving quantitative values of the experimental part in the mine studied. I think that broader and more general conclusions should be added that are useful for other places.

5. Regarding the role played by the presence of water, little is said in results and discussion. I think that this aspect should be reinforced. Also in the conclusions should be treated more widely.

Kind regards

Author Response

Response to Reviewer 1 Comments

Thanks for your comments and suggestions concerning our paper entitled”The effects of backfill mining on strata movement rule and water inrush: a case study”.Those comments are all valuable and very helpful for revising and improving our paper, as well as the important guiding significance to our researches.

We completely accept your opinion and have made modification.

Point 1:To guide the reader, in the final part of theintroduction should be explicitly stated and more clearly what are theobjectives of the work. Next, a brief description of the content of thearticle, its parts or sections with a brief description should be made.

Response 1:In final part of introduction, objectives of the word was described. A brief description of content and its parts was added according to suggestions.

Point 2: In the study area section the geological structurein the immediate vicinity of the mine is clearly described, but I believe thata more general vision should be provided, at a regional scale, including geological map with stratigraphic units or geological formations, tectonicstructure with structural directions and dips, and lithological description. Across section on a regional scale should also accompany this figure.

Response 2:A new geological map include tectonic structure with structural directions and dips was increased. So more information could be got with the map.

Point 3:In general, the work is very focused as a casestudy. In my opinion, it should be given a broader meaning. For example, inResults and discussion section, apart to commenting onthe specific quantitative data for the Caozhuang Coal Mine, more general,qualitative comments should be made and the repercussions that may be usefulfor places other than this mine should be commented.

Response 3:According to the suggestion, a broader comments was added in results and discussion. Differences between backfilling mining and traditional longwall mining was analyzed qualitively. More significance of backfill mining can be acquired in the word.

Point 4:With the conclusions the same thing happens, in myopinion they are too poor since they are limited to giving quantitative valuesof the experimental part in the mine studied. I think that broader and moregeneral conclusions should be added that are useful for other places.

Response 4:Some abundant comments was added in terms of strata movement and water inrush in the conclusions.

Point 5:Regarding the role played by the presence of water,little is said in results and discussion. I think that this aspect should bereinforced. Also in the conclusions should be treated more widely.

Response 5:Meaning of backfill mining to water inrush was reinforced, a more wide comments was described in conclusions.

Thank you again for your review.

Best wishes.

Mr. Jiahui Lin

E-Mail: [email protected]

Reviewer 2 Report

The paper has presented a case study on the effects of backfill in mining. Therefore, the novelty is low, but it is very interesting to engineering readers and may provide some limited guidance for engineering practice. The data is collected experimentally and there is no further deep data analysis, so the scientific soundness is limited to some extent. 

Some essential works have been overlooked in the literature review. Here are some suggestions about stress distribution in backfill:

SIVAKUGAN, N., WIDISINGHE, S. & WANG, V. Z. 2013. Vertical stress determination within backfilled mine stopes. International Journal of Geomechanics, 14, 06014011.

LIU, G., LI, L., YANG, X. & GUO, L. 2016. Numerical analysis of stress distribution in backfilled stopes considering interfaces between the backfill and rock walls. International Journal of Geomechanics, 17, 06016014.

TO, P., & SIVAKUGAN, N. 2018. Boundary stress distribution in silos filled with granular materials. 9th European Conference on Numerical Method in Geotechnical Engineering, Vol. 2, 863-868

Some minor comments

Line 38: " the overburden'  s weight" shoul be "overburden stress".

Line 43: "However, few" should be "However, only few"

Line 51: "and is abundant in" should be "and abundance of"

Line 67: "The water-inrushing" should be "Hence, the water-inrushing"

Line 79: "is composed of" should be "comprises"

Line 100: "is composed of" should be "has"

Line 119: "As the roof compressed." should be "As the roof compressed,"

Line 150: "vale and the" should be "valve, and the". How can water be injected through the pressure gauge?

Line 177:  "working face advances" should be " working face advanced". Confusing structure.

Line 206: "RFD is 104 mm" should be "RFD was 104 mm".

Line 223: "Backfill stresses monitored" should be "Backfill stress was monitored"

Line 244: "didn't" should be "did not"

Line 254: font changed.

Line 279: ", and therefore FFD" should be "and, therefore, FFD"

Author Response

Response to Reviewer 2 Comments

Thanks for your comments and suggestions concerning our paper entitledThe effects of backfill mining on strata movement rule and water inrush: a case study.Those comments are all valuable and very helpful for revising and improving our paper, as well as the important guiding significance to our researches.

We completely accept your opinion and have made modification.

Point 1:Some essential works have been overlooked in the literature review.

Response 1:I am sorry for overlooking the essential works done by our predecessors in the literature review, which is indeed my negligence in work.I have studied these important works and added them into the literature review of my paper.

Point 2:Some minor comments.

Response 2:I also made a lot of mistakes in words, punctuation and fonts, which shows that I need to be more careful and improve my English.Thanks for pointing it out to me .I have finished the modification and the modifications are as follows.

Line 39: I changed " the overburden'  s weight" to "overburden stress".

Line 46: I changed "However, few" to "However, only few"

Line 56: I changed "and is abundant in" to "and abundance of"

Line 74: I changed "The water-inrushing" to "Hence, the water-inrushing"

Line 86: I changed "is composed of"to "comprises"

Line 109: I changed "is composed of " to "has"

Line 129: I changed"As the roof compressed." to "As the roof compressed,"

Line 186I changed "working face advances" to " working face advanced".

Line 217: I changed "RFD is 104 mm" to "RFD was 104 mm".

Line 234: I changed "Backfill stresses monitored" should be "Backfill stress was monitored"

Line 255: I changed "didn't" to "did not"

Line 259: Font changed.

Line 285: I changed ", and therefore FFD" to "and, therefore, FFD"

Thank you again for your review.

Best wishes.

Mr. Jiahui Lin

E-Mail:[email protected]

Round 2

Reviewer 1 Report

Dear Authors and Editor,

The authors have put diligent efforts in responding to reviewers’ comments and they are revised the paper accordingly. The paper is ready. I am happy to recommend acceptance of the paper.

Best regards

Reviewer 2 Report

The authors have provided sufficient amendment. However, big proofreading work with moderate English changes is required.